# Characterization of the Uniformity of High-Flux CdZnTe Material

**DOI:** 10.3390/s20102747

**Published:** 2020-05-12

**Authors:** Matthew Charles Veale, Paul Booker, Simon Cross, Matthew David Hart, Lydia Jowitt, John Lipp, Andreas Schneider, Paul Seller, Rhian Mair Wheater, Matthew David Wilson, Conny Christoffer Tobias Hansson, Krzysztof Iniewski, Pramodha Marthandam, Georgios Prekas

**Affiliations:** 1Rutherford Appleton Laboratory, UKRI Science & Technology Facilities Council, Oxon OX11 0QX, UK; paul.booker@stfc.ac.uk (P.B.); simon.cross@stfc.ac.uk (S.C.); matthew.hart@stfc.ac.uk (M.D.H.); lydia.jowitt@stfc.ac.uk (L.J.); john.lipp@stfc.ac.uk (J.L.); andreas.schneider@stfc.ac.uk (A.S.); paul.seller@stfc.ac.uk (P.S.); rhian-mair.wheater@stfc.ac.uk (R.M.W.); matt.wilson@stfc.ac.uk (M.D.W.); 2Redlen Technologies, Saanichton, BC V8M 1X6, Canada; conny-h@slac.stanford.edu (C.C.T.H.); kris.iniewski@redlen.com (K.I.); pram.marthandam@redlen.com (P.M.); g.prekas@live.co.uk (G.P.)

**Keywords:** CdZnTe, pixel detector, X-ray detector, X-ray spectroscopy, X-ray imaging, high flux

## Abstract

Since the late 2000s, the availability of high-quality cadmium zinc telluride (CdZnTe) has greatly increased. The excellent spectroscopic performance of this material has enabled the development of detectors with volumes exceeding 1 cm^3^ for use in the detection of nuclear materials. CdZnTe is also of great interest to the photon science community for applications in X-ray imaging cameras at synchrotron light sources and free electron lasers. Historically, spatial variations in the crystal properties and temporal instabilities under high-intensity irradiation has limited the use of CdZnTe detectors in these applications. Recently, Redlen Technologies have developed high-flux-capable CdZnTe material (HF-CdZnTe), which promises improved spatial and temporal stability. In this paper, the results of the characterization of 10 HF-CdZnTe detectors with dimensions of 20.35 mm × 20.45 mm × 2.00 mm are presented. Each sensor has 80 × 80 pixels on a 250-μm pitch and were flip-chip-bonded to the STFC HEXITEC ASIC. These devices show excellent spectroscopic performance at room temperature, with an average Full Width at Half Maximum (FWHM) of 0.83 keV measured at 59.54 keV. The effect of tellurium inclusions in these devices was found to be negligible; however, some detectors did show significant concentrations of scratches and dislocation walls. An investigation of the detector stability over 12 h of continuous operation showed negligible changes in performance.

## 1. Introduction

Cadmium zinc telluride (CdZnTe) is a compound semiconductor that has been studied for many years for its application in the detection of radiation, specifically X-rays and γ-rays. The advantage of this material over traditional direct conversion detector materials like silicon and germanium is its high density (~5.8 g cm^−3^) and high resistivity (~10^11^ Ω cm), which make it ideally suited for the detection of high energy X-rays and γ-rays without the need for cryogenic cooling. The resistivity of CdZnTe also exceeds that of CdTe, enabling the production of thicker detectors (>1 mm) that are needed to stop photons with energies >50 keV [1].

One of the longest standing challenges that has faced CdZnTe detectors is their performance under irradiation with very high photon fluxes (>10^6^ ph s^−1^ mm^−2^). Under these conditions, detectors have been prone to the polarization phenomenon in which the electrical field in the detector is modified over time due to a buildup of trapped charge in the crystal [2]. Historically, this has limited the use of CdZnTe detectors to application areas, such as nuclear material detection, where photon fluxes are much lower [3]. In order for CdZnTe detectors to be of use in high photon flux applications like Computed Tomography (CT) imaging and beam line instrumentation at synchrotrons and X-ray free electron lasers (XFELs), the polarization phenomenon needs to be suppressed or removed completely.

Driven by the needs of the medical and security imaging communities, Redlen Technologies has developed a new ‘high-flux-capable’ grade of CdZnTe material [4], referred to here as HF-CdZnTe. Results from this material were first reported by Thomas et al. in 2017 [5] and it has since gone on to be tested at intense light sources like the LCLS XFEL [6] and at the ESRF synchrotron [7]. These measurements have demonstrated how this new HF-CdZnTe can operate at fluxes > 10^6^ ph s^−1^ mm^−2^, with no evidence observed for the onset of the polarization phenomenon. These successful results suggest that, for the first time, the widespread use of CdZnTe-based detectors for high flux applications may become a reality.

One of the key areas of this material’s performance that is still to be studied is the repeatability of results across multiple detectors with a reasonable detection area (> 2 cm^2^). Previous measurements with large-area CdZnTe and CdTe detectors have demonstrated spatial variations in the detector response due to crystalline defects, such as tellurium inclusions, or due to local variations in the electric field. If HF-CdZnTe devices are going to be suitable for imaging applications, then it is a prerequisite that large detection areas with a high degree of uniformity can be produced [8,9]. In this paper, the spatial uniformity of 10 2.0-mm-thick HF-CdZnTe detectors were characterized using the High Energy X-ray Imaging Technology (HEXITEC) spectroscopic imaging Application Specific Integrated Circuit (ASIC) [10] operating at a flux of <10^4^ ph s^−1^ mm^−2^. The spectroscopic information produced by the HEXITEC ASIC allows the spatial uniformity of the material to be quantified at sub-mm scales using key metrics, such as the pixel to pixel variations in the energy resolution and the number of counts.

## 2. Materials and Methods

In this paper, the results are reported for the characterization of a sample of 10 pixelated HF-CdZnTe detectors, each with an area of 4.2 cm^2^, which were fabricated by Redlen Technologies and are read out with the Science and Technology Facitlities Councils’s (STFC) HEXITEC ASIC [10]. Details on the individual components used to make these measurements are detailed below.

### 2.1. High Flux CdZnTe Detectors

In the commonly used spectroscopic-grade CdZnTe material, the growth process is optimized to deliver crystals with excellent electron charge transport properties, which is ideal for thick sensors (>5 mm) with large pixels (>1 mm) that are typically used for γ-ray spectroscopy applications [3]. The disadvantage of this material is that the hole carrier lifetime is normally very short (τ_h_ ~ 0.2 μs) and leads to significant numbers of carriers being trapped. At high photon fluxes, the build-up of these trapped holes result in the polarization phenomenon. In this HF-CdZnTe material, a different approach is taken where the hole lifetime is optimized, leading to an increase of an order of magnitude (τ_h_ ~ 2.0 μs) [5]. The increase in the hole lifetime enables detectors produced from HF-CdZnTe material to operate at the higher photon fluxes used in many photon science applications.

The HF-CdZnTe material used in this study was grown by Redlen Technologies using their proprietary THM growth process [4]. Redlen were also responsible for the processing and fabrication of the sensor electrodes for these detectors. The detectors consist of an array of 80 × 80 pixels on a 250-μm pitch with a pad size of 200 μm × 200 μm and a gap size of 25 μm surrounded by a 100-μm-wide guard ring along three edges and 200 μm along one edge. Platinum electrodes are used for both the pixelated anode and the planar cathode. The total sensor dimensions are 20.35 mm × 20.45 mm × 2.00 mm. An example of an optical image of the bare HF-CdZnTe sensor can be seen in Figure 1a. Individual HF-CdZnTe sensors are hybridized to a HEXITEC ASIC using the STFC’s interconnect facilities [11]. Bumps of silver-loaded epoxy are deposited on each pixel of the sensor using a stencil printing technique; typically, bumps have diameters of ~ 120 μm and heights of 30 μm. An image of a portion of a sensor following the deposition of the silver epoxy bumps can be seen in the inset in Figure 1a. In a separate process, a gold stud is produced on each of the pixel inputs of the HEXITEC ASIC each with a diameter of 50 μm and a height of ~ 30 μm. The sensor and ASIC are then flip-chip-bonded together and cured in situ at 150 °C. The hybrid is then attached to the aluminum carrier and the ASIC I/O pads aluminum wedge-bonded to the PCB. A fully assembled HF-CdZnTe HEXITEC module can be seen in Figure 1b.

### 2.2. The HEXITEC Detector System 

The HEXITEC ASIC is a spectroscopic X-ray imaging readout chip that has been used across a broad range of application areas, including materials science [12,13], solar physics [14], and laser physics [15]. The HEXITEC ASIC consists of 80 × 80 channels on a 250-μm pitch. Each of these pixels consists of a charge-sensitive preamplifier, a shaping amplifier, and a peak-track-and-hold circuit [10]. The analogue outputs of the ASIC are read out using a roller-shutter scheme. The magnitude of the voltage read from the peak-track-and-hold circuit is directly proportional to the energy of the photon that is stopped in the CdZnTe sensor. As the ASIC was designed for the readout of high-Z compound semiconductors, it is only sensitive to the electron signals generated in the CdZnTe sensor.

Each HEXITEC module, see Figure 1b, is mounted in a HEXITEC ‘GigE’ data acquisition system (DAQ) [16]. The DAQ contains the entirety of the readout electronics, including the analogue-to-digital converters (ADCs) that digitize the analogue voltages readout from the chip. For all measurements, the GigE DAQ system was operated at a frame rate of 1.6 kHz. The DAQ also delivers the high-voltage supply that biases the CdZnTe sensor as well as the temperature control system that maintains the ASIC at room temperature (28 °C). Unless stated otherwise, the measurements presented in this paper were collected with a bias voltage of 750 V (325 V mm^−1^). The higher resistivity of the CdZnTe material means that, unlike CdTe detectors, there was no need to use periodic bias cycling, which is typically used in CdTe to avoid bias-induced polarization.

### 2.3. Energy Calibration Procedure

In order to study the performance of the HF-CdZnTe sensors, each of the detector modules requires an energy calibration, which is completed using the known emission lines of an ^241^Am-sealed source. The peaks used included the 13.94, 17.75, 26.34, 36.37, and 59.54 keV lines. It should be noted that 36.37 keV is not an emission of the sealed source but an escape peak that is generated due to the creation of a 23.17 keV cadmium fluorescence X-ray in the CdZnTe sensor that subsequently escapes the sensor volume.

Each of the detectors were irradiated with a 183 MBq ^241^Am sealed source positioned ~15 cm to ensure a flat field across the entire 4 cm^2^ of the detector. Data were collected for 300 s for each detector, with ~10^5^ events recorded per pixel (3 × 10^3^ ph s^−1^ mm^−2^). At this distance, the occupancy of a single frame of data, Figure 2a, was <5% of pixels, which allows a charge sharing discrimination (CSD) algorithm to be used effectively [17]. Using CSD only events in a frame of data that occurs in a single pixel (i.e., no charge observed in immediate neighbor pixels) is included in the analysis. Using bespoke code written in Matlab R2019b, each frame is inspected in turn for charge sharing events and per-pixel spectra are compiled across the entire data set (~10^6^ frames), only including single pixel events. The effect of CSD on a typical single pixel spectrum can be seen in Figure 2b, where a continuous low-energy background is removed, making it easier to identify the individual calibration peaks. The CSD spectrum for each pixel is inspected and the location of the calibration peaks identified. The position of these peaks is then used to perform a linear calibration of each pixel, extracting a gradient and intercept for each channel as shown in Figure 3. The average values of the gradient and intercepts measured for these detectors were 29 eV ADU^−1^ and 790 eV, respectively. The most visible variations are in the values of the gradients, Figure 3b, which show four distinct regions of 20 × 80 pixels. These variations are not due to the sensor material but are a result of small variations in the response of the 4 ADCs in the HEXITEC DAQ system, each of which reads out a block of 20 × 80 pixels.

In parallel to the energy calibration, the low energy threshold for each pixel is also calculated. For this analysis, the low energy threshold was calculated by first determining the position of the noise peak at low channel numbers (< 100 ADU) and then determining its width. The pixel threshold was then defined as the position 8σ away from the noise edge. Figure 3d shows the magnitude and spatial distribution of the threshold values for a typical detector. The average threshold per pixel was of the order of 2 keV, with pixels at the sensor edge displaying higher values of the order 3–5 keV.

## 3. Results

With the energy calibrated for each of the detector modules, the data sets were reprocessed by this time applying the per-pixel energy calibration calculated as described in Section 2.3. The calibrated data was then used to extract key indicators of the detector’s performance that included the energy resolution of the individual pixels, the percentage of interactions that displayed charge sharing, and if there were any crystalline defects present in the sensor volume. Each of these performance indicators are described below.

### 3.1. Spectroscopic Performance

An important indicator of the performance of compound semiconductor materials, such as CdZnTe, is their energy resolution, quantified here using the full width at half maximum (FWHM) of the main photo-peak. The energy resolution is dependent on a number of factors, including the charge transport properties of the material, the charge carrier mobility (μ), and the carrier lifetime (τ), as well as the detector geometry and electronics. The magnitude of spatial variations in the energy resolution also provides an excellent indicator of the uniformity of the HF-CdZnTe material.

The ultimate limit of the energy resolution of direct conversion detectors is described by the Fano-limited energy resolution. When an X-ray is absorbed in the semiconductor, the processes that give rise to individual charge carriers are not independent events. The band structure of the material limits the numbers of ways that the semiconductor may be ionized and, as a result, the energy resolution that is achievable is better than that predicted by simple statistical considerations. The Fano-limited energy resolution (Δ*E_Fano_*) is described by Equation (1):(1)$$\Delta E_{Fano}=2.35\sqrt{FWE_{\gamma }},$$
where *F* is the Fano factor and *W* is the electron hole pair creation energy; these have values of 0.1 and 4.62 eV ehp-1 in CdZnTe material [18]. In recent years, the spectroscopic performance of CdZnTe and CdTe detectors has improved dramatically, with FWHM measured at 59.54 keV of the order 1–2 keV [9,16,19,20], compared to the theoretical Fano-limited minimum of 0.39 keV.

The spectroscopic performance of the HF-CdZnTe material was evaluated using the ^241^Am sealed source as shown in Figure 4, where each channel of the spectra was 200 eV wide. The charge sharing-discriminated ^241^Am γ-ray spectrum of a single pixel is compared to that of the global spectrum (all the channels summed together) in Figure 4a; Figure 4b shows how the distribution of FWHM values for 59.54 keV photo-peak across the entire detector. The average value of the FWHM across all 6400 pixels of this detector was found to be 0.79 keV with a standard deviation in the values of ±0.15 keV. In the best performing pixels, the FWHM was as low as 0.54 keV, which is approaching the noise limitations of the HEXITEC ASIC (~0.50 keV). For the lowest energy line of the sealed source, 13.94 keV, the FWHM was measured to be 0.61 keV with a standard deviation in the values of ±0.13 keV. These FWHM values are some of the best measured with CdZnTe material and compare favorably to the Fano-limited energy resolution of 0.39 keV as well as measurements by other groups using Acrorad CdTe detectors that have long been considered the gold standard within the community [9,19,20]. As seen in the map of the spatial variation in the FWHM values, the detector also shows a high degree of uniformity, with 99.93% of pixels having an FWHM of less than 2 keV. The poorest performing pixels for this detector were found at the edge of the sensor and are consistent with the increased leakage current and uncorrected charge sharing that occurs in these regions [21].

The spectroscopic performance of all 10 of the HF-CdZnTe detectors were evaluated using the same method and the results are summarized in Table 1. Across the sample of 10 detectors, an average FWHM of 0.83 keV was measured, which is comparable to the best energy resolution previously measured with the Acrorad Schottky-anode CdTe using the HEXITEC ASIC [16]. It is worth noting that one of the key advantages of the CdZnTe is that, unlike the CdTe, it does not require the bias voltage to be cycled periodically to avoid bias-induced polarization.

### 3.2. Charge Sharing

During data processing, each frame of data was inspected and events that displayed charge sharing identified. Figure 5a shows the distribution of the number of pixels involved in interactions, the event multiplicity, measured for sensor D185735. The magnitude of the number of shared events is a result of the detector geometry, the energy of the incoming photon, and the bias voltage applied across the detector [17]. The geometric contribution to the overall charge sharing was described by Iniewski [22] and predicts that 34.8% of the interactions should experience sharing under these operating conditions. In reality, a charge sharing percentage of 55.9% was measured for the detector in Figure 5a, and across the sample of 10 detectors the average was 58.0%, see Table 1. The difference between the prediction and the measurement is likely to be due to the generation of cadmium and tellurium X-ray fluorescence photons during interactions. These fluorescence photons have ranges in the detector of 60–120 μm in the CdZnTe, which are comparable to the pixel size. When generated, the Cd and Te XRF photons are emitted in 4π and have a high probability of stopping in the neighboring pixels, leading to an increase in the number of shared events relative to the prediction from the detector geometry alone. It is also worth noting that there is very little spatial variation in the charge sharing percentage across the entire detector, see Figure 5a, with only major variations observed at the physical edge of the sensor. The spatial uniformity of the charge sharing suggests that the electric field across the detector is also very uniform and does not contain the electric field variations that have been observed previously in spectroscopic-grade CdZnTe material from Redlen [23].

As the HEXITEC system records each frame of data, any charge sharing corrections are applied during data processing, allowing different algorithms to be used. Figure 5b shows the global spectra from detector D185735 that is produced using three different techniques. Without any correction, the spectrum sits upon a broad distribution of lower energy events that are a result of the sharing of charge between multiple pixels that erroneously appear as separate events. The lower energy background is removed in the charge sharing discriminated spectrum that only includes the events detected in a single pixel (m = 1). The use of a charge sharing discrimination algorithm is effective in producing the highest resolution spectroscopy but at a cost of 55.9% of the total counts. An alternative is to use a charge sharing addition algorithm where the charge in neighboring pixels are summed together in an attempt to recover the original energy of the event. In Figure 5b, the reconstructed spectrum is shown for those events that were shared between two pixels (m = 2). The charge sharing addition reconstructed spectrum is much broader than that of the single pixel events, with a Gaussian fit of the photo-peak yielding an. FWHM of 1.7 and 0.7 keV, respectively. The broadening cannot merely be explained by the addition of noise sources; instead, it is due to the loss of charge that occurs between pixels. The charge loss is a result of a lower field region between pixels that results in additional charge trapping. This phenomenon has been studied previously in HEXITEC devices and is beyond the scope of this paper [24,25,26].

### 3.3. Detector Defects

Exceptional spectroscopic performance was measured in each of the 10 HF-CdZnTe detectors; however, some pixel to pixel variation was observed in the number of counts detected. Figure 6 compares the intensity maps produced across all 10 detectors when exposed to a flat field exposure with the 59.54 keV emission of the ^241^Am-sealed source. While some of the detectors show little to no defects, notably sensor D185735, a number do show areas of considerable variation. The uniformity of these detectors can be quantified by calculating the variation in the number of counts in the 59.54 keV peak per pixel (σ_counts_), see Table 1. The best performing detectors show pixel to pixel variations of 5.5% (D185735) while the worst shows 12.2% (D185740). The uniformity of the best HF-CdZnTe detectors compares well to that of Schottky CdTe detectors, which have previously been tested with HEXITEC that also show pixel to pixel variations of ~5% [16]. In the rest of this section, the different types of non-uniformity observed in the HF-CdZnTe detectors will be examined in more detail.

Figure 7a compares an optical image of the pixelated anode of sensor D180265 taken prior to hybridization to the measured X-ray response, Figure 7b. Of all of the 10 devices tested, D180265 had the poorest bond yield, with 4.48% of pixels having little to no spectroscopic performance. As can clearly be seen in the X-ray image, the majority of these poor pixels are in one corner of the detector and this appears to be consistent with an area of increased reflectance in the optical image of the surface. The area of low bond yield may be due to poor surface flatness in this region of the sensor that results in the silver bumps and gold studs on the ASIC failing to mate.

A number of features in the optical image shown in Figure 7a can be correlated with the X-ray image in Figure 7b. These defects include a deep scratch across a block of 3 × 4 pixels that results in a region of little to no X-ray response corresponding to the scratched pixels. As clear damage can be seen to the electrode material, it is not surprising that this results in a large reduction of the number of counts in these pixels. A number of other large features seen in the right-hand side of the optical image in Figure 7 can also be correlated with the X-ray response. While at first these defects may appear to be scratches of the electrode, closer inspection of both images demonstrates that these features do not perfectly align. The optical image also suggests the feature is either a scratch to the sub-electrode CdZnTe surface or some other crystalline defect. It is also notable that these features display local increases in the measured intensity rather than the decrease observed in the large scratch that pierces the electrode; this suggests that these features are influencing the local electric field.

The same features are also observed in some of the other HF-CdZnTe devices, see Figure 6, but the highest densities are in detectors D185740 and D185749. Figure 8 compares the spatial variation in the counts measured to the energy resolution (FWHM) and charge sharing percentage for each of the pixels. These measurements demonstrate, at least at this pixel pitch, that the energy resolution of a pixel is independent of the density of these features. However, a correlation with the magnitude of the charge sharing experienced by a pixel and these defects can be observed. Pixels showing higher than average counts in the intensity maps also show less charge sharing; this suggests that these features are causing local variations in the electric field that increase the effective collection volume of the pixels that are coincident with the defects.

Similar features in CdZnTe material have been reported previously by Veale et al. [27] in Redlen spectroscopic-grade CdZnTe and more recently by Tsigaridas et al. [7] in Redlen HF-CdZnTe. Similar defects have also been studied in other material systems, such as CdTe [28] and GaAs [29], where they have been described as ‘dislocation walls’. These defects are formed due to the stresses the crystal experiences during growth [30] and the distribution of these dislocation networks result in local variations in both the photo- and dark-currents in the sensor. While the presence of these defects is not ideal, they are static and the use of flat fielding is effective in removing the effect they have on their imaging performance [7,29]. The results from this sample of 10 detectors suggest that both scratching of the sub-electrode CdZnTe surface and dislocation walls plays a role in the performance of this HF-CdZnTe material. 

One of the other defect types that have historically had a large impact on the performance of CdZnTe and CdTe detectors, and as a result have been the focus of intense study, are tellurium inclusions [31]. Of the 10 detectors characterized in this study, only one, D185734, showed evidence for inclusions of appreciable size, suggesting that the concentration is low in the HF-CdZnTe material. Figure 9 compares the spectrum measured in a pixel that is suspected to contain an inclusion within its volume and the spectrum from an unaffected pixel in its vicinity. The effect of the inclusion is two-fold: First, there is an impact on the spectroscopic performance of the pixel due to significant low-energy tailing caused by charge trapping and the second is an increase in the frequency of charge sharing. These finding suggests that not only are impurities in and around the inclusion resulting in charge trapping but also that its presence is leading to a local reduction in the electric field and, as a result, an increase in the proportion of charge sharing. These effects are consistent with previous results measured in CdZnTe and CdTe detectors [27,31].

### 3.4. Temporal Stability

Another important consideration in accessing the performance of the HF-CdZnTe material is its temporal stability. For techniques like computed tomography that are widely used for materials science, security, and medical imaging, detectors may be exposed to high fluxes of X-rays for many hours at a time and must remain stable for the duration of the measurements. Temporal variations in the performance of CdTe and CdZnTe detectors have commonly been reported. These phenomena are typically caused by flux-induced [2,32] and bias-induced polarization [33,34] in which a buildup of space charge, caused by the trapping of carriers, leads to a progressive modification of the electric field. These changes in electric field result in a reduction in both the counting efficiency and the spectroscopic performance of detectors. The rates of these changes are dependent on a number of factors, such as the photon flux, detector electrode type, the bias voltage, and temperature.

In the case of bias-induced polarization, the use of a blocking (Schottky) electrode prevents carriers leaving the sensor, resulting in the formation of space charge under the electrode independent of the photon flux. The presence of bias-induced polarization is normally addressed through the use of a bias refresh scheme, where the detector bias is periodically set to 0 V for a short time of the order of a second in order to allow the space charge to recombine. The period of bias refreshing can be on the order of minutes or hours depending on the detector conditions [16,33,34,35]. In the case of flux-induced polarization, the large number of photo-generated carriers results in excessive numbers of trapped holes that form the space charge within the sensor. Once polarization occurs, only a large reduction or removal of the photon flux will allow the detector to recover [2,32]. 

Recent encouraging results from the characterization of HF-CdZnTe material coupled to single photon counting [7] and integrating detector ASICs [6] have demonstrated that the material is capable of operating at high photon fluxes, with negligible changes detected in detector performance. These results suggest that the polarization phenomenon in this material is reduced or suppressed. The advantage of the spectroscopic information provided by the HEXITEC ASIC is that it allows any small changes in the detector material under extended operation to be studied; it is worth noting that HEXITEC is limited to use for low X-ray fluxes (<10^5^ ph s^−1^ mm^−2^). In order to study the detector stability in more detail, data was collected for 300 s at 900-s intervals for a total of 9–12 h with detector D185735. Measurements were made with the detector temperature fixed at the standard 28 °C operating temperature as well as at a reduced temperature of 18 °C. A constant bias of 750 V (325 V mm^−1^) was applied to the detector and no bias refresh scheme was used. Figure 10 shows the evolution of the global spectrum at 28 °C over the course of a 9-h exposure. The change in the number of counts, position and width of the 59.54 keV photo-peak, as well as the percentage of charge sharing events were evaluated for each acquisition, see Figure 11.

At both of the operating temperatures measured, small changes in the overall performance of the detector were observable. For example, the average FWHM measured in the detector over 9 h was found to change from 770 to 840 eV at 28 °C and 760 to 780 eV at 18 °C. These levels of changes are negligible and show no clear spatial correlation within the detector. As with previous studies at higher photon fluxes, this suggests that the response of the HF-CdZnTe material shows a high degree of stability over many hours, which is more than sufficient for the typical operation conditions of these detectors.

## 4. Conclusions

A set of 10 HF-CdZnTe detectors grown and processed by Redlen Technologies were assembled and characterized using the HEXITEC ASIC by UKRI STFC. The characterization of these detectors has demonstrated the exceptional spectroscopic performance of the HF-CdZnTe material, with energy resolutions of <900 eV measured at 59.54 keV across all 10 devices. The spatial uniformity of the best performing detectors was comparable to that of Acrorad Schottky CdTe detectors, with pixel to pixel variations in the number of counts of <6%. In some of the detectors, the spatial uniformity of the detectors was poorer, with pixel to pixel variations in the number of counts detected of >10%. These detectors showed evidence for a high concentration of scratching of the CdZnTe crystal surface prior to electrode deposition as well as the presence of crystalline defects, such as dislocation walls. These defects lead to local variations in the counting performance and the proportion of events that demonstrate charge sharing; however, this had little effect on the energy resolution of individual pixels. Measurements also showed that 9 out of the 10 detectors were free of large inclusions, suggesting that the overall inclusion concentration is low in the HF-CdZnTe material. An investigation of the detector response over a 12-h period under constant bias showed negligible changes in performance, providing further evidence for the excellent temporal stability of this material. These results suggest that HF-CdZnTe material is an excellent candidate for the production of a new generation of X-ray imaging cameras at synchrotrons, FELs, and beyond. 

## Figures and Tables

**Figure 1 sensors-20-02747-f001:**
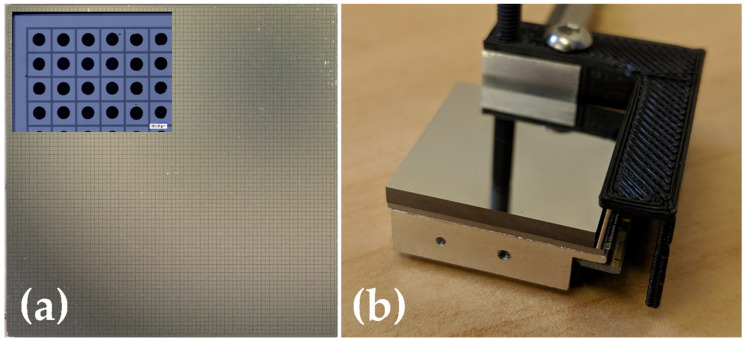
Optical images of a 2-mm-thick HF-CdZnTe HEXITEC detector: (**a**) An image of the 80 × 80 array of 250 μm pitch pixels; (inset) a higher magnification image of a 1.25 mm × 1.75 mm region of the detector after the deposition of silver-loaded epoxy bumps during hybridization; (**b**) an image of the final detector module after hybridization of the CdZnTe and HEXITEC ASIC.

**Figure 2 sensors-20-02747-f002:**
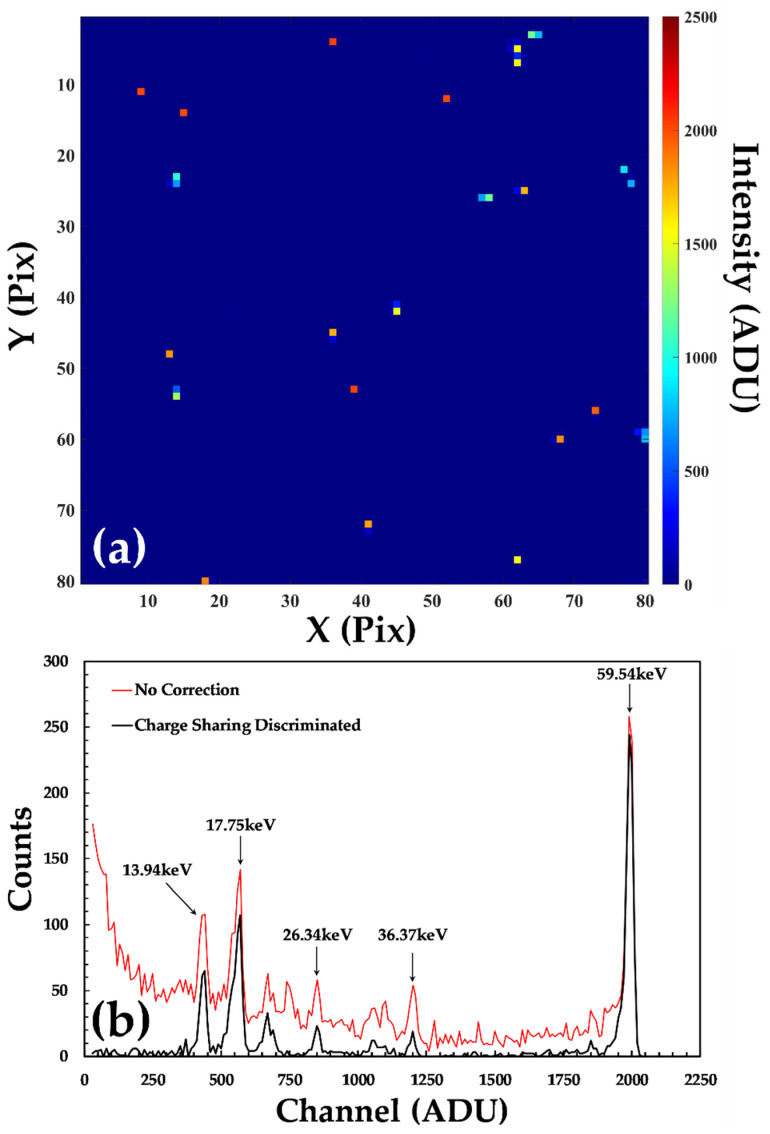
Examples of the data collected with the HF-CdZnTe HEXITEC detector D185735: (**a**) a single frame of data captured with the detector system; (**b**) an example of the un-calibrated spectrum recorded with a single pixel. The raw spectrum is shown in red while the spectrum after the removal of charge sharing events is shown in black.

**Figure 3 sensors-20-02747-f003:**
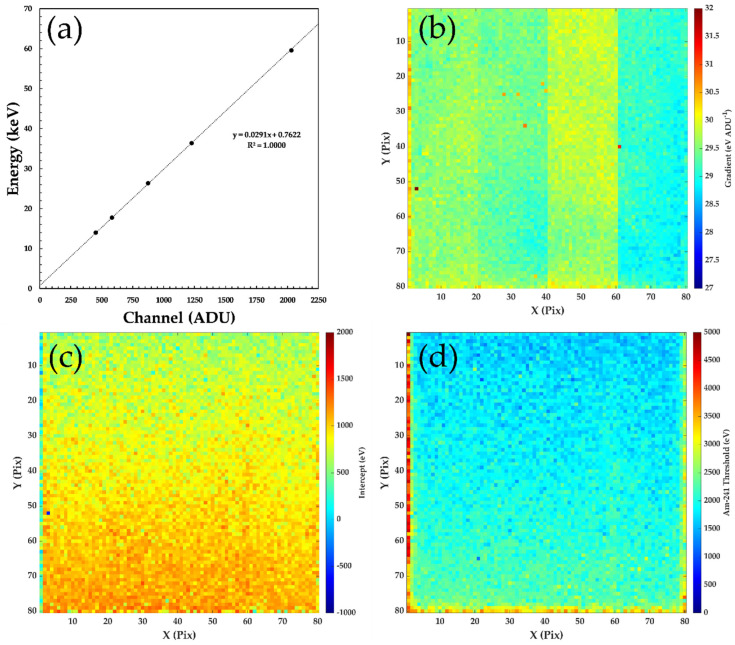
An example of the calibration of an 80 × 80 pixel HF-CdZnTe HEXITEC detector D185735 (**a**) the linear calibration of a single pixel of the detector using the known emissions of the ^241^Am γ-ray sealed source; (**b**) the gradient coefficient extracted for each pixel of the array; (**c**) the intercept coefficient extracted for each pixel of the array, and (**d**) the low energy threshold calculated for each of the pixels.

**Figure 4 sensors-20-02747-f004:**
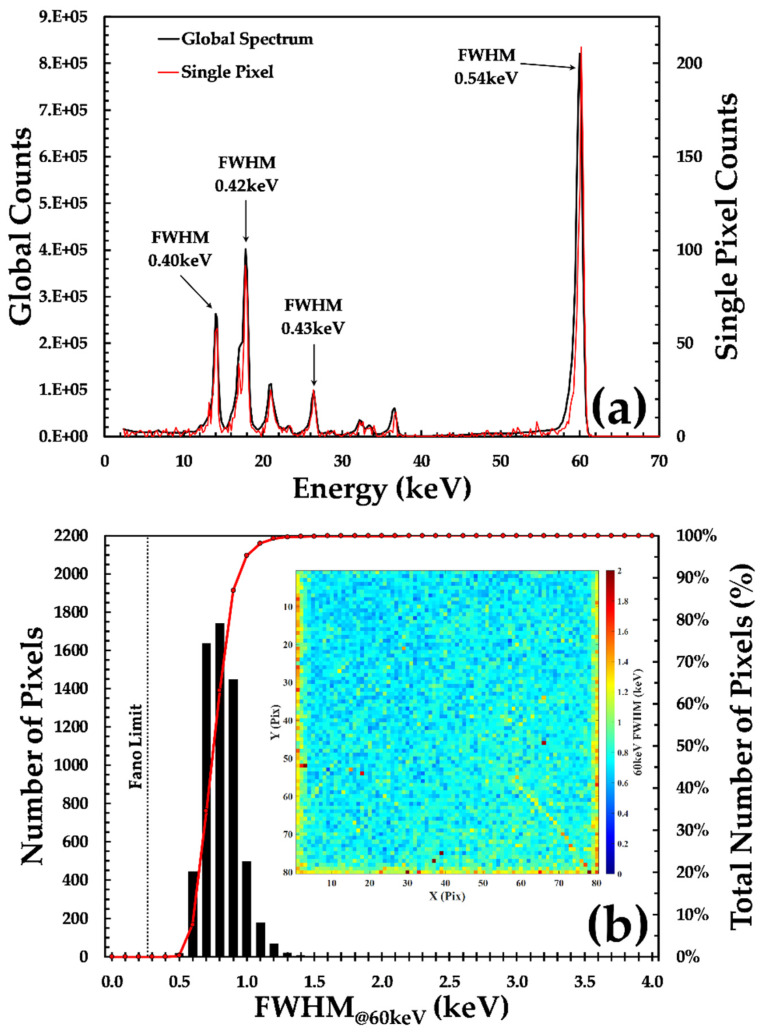
The spectroscopic performance of the HF-CdZnTe HEXITEC detector D185735 (**a**) the ^241^Am γ-ray spectrum after energy calibration and the removal of charge sharing events, a single pixel spectrum is shown in red while the global spectrum is shown in black; (**b**) a histogram of the measured FWHM of the ^241^Am γ-ray across the entire detector array; (inset) the spatial distribution of the FWHM values.

**Figure 5 sensors-20-02747-f005:**
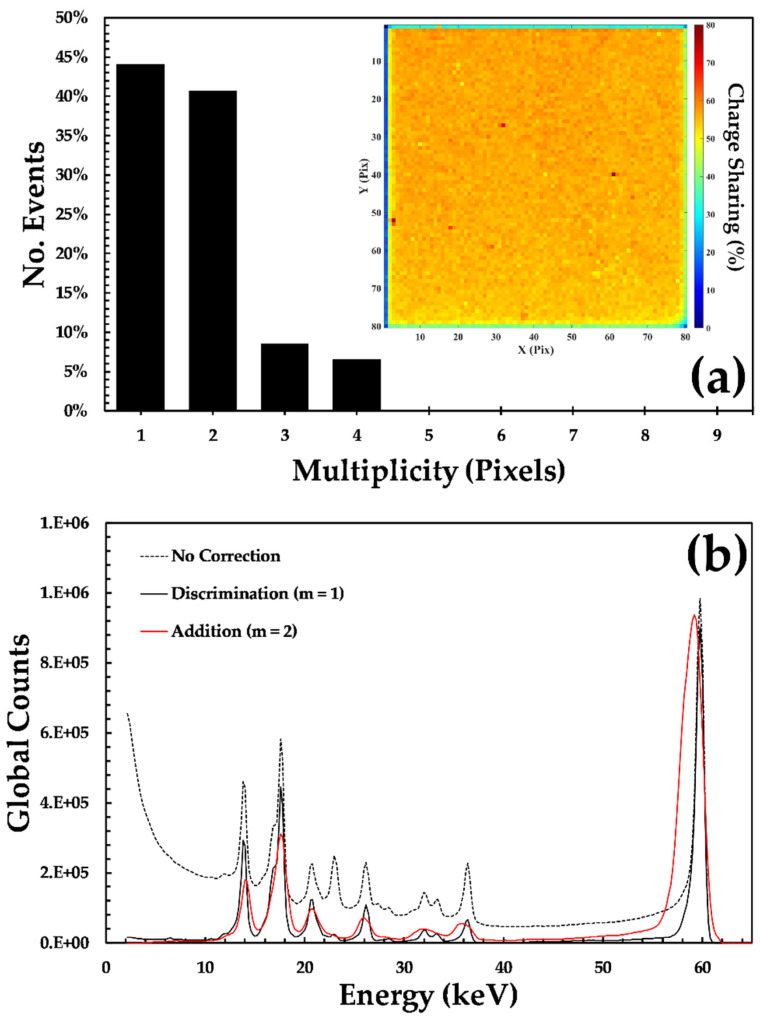
The role of charge sharing in the performance of a HF-CdZnTe detector; (**a**) a histogram showing the distribution of the number of pixels, the multiplicity, involved in an interaction; (**inset**) a map indicating the spatial variation in the percentage charge sharing per pixel; (**b**) the global spectrum without correction (dashed black line), with all charge sharing events removed (black solid line) and the spectrum of the m = 2 events after charge sharing addition (solid red line).

**Figure 6 sensors-20-02747-f006:**
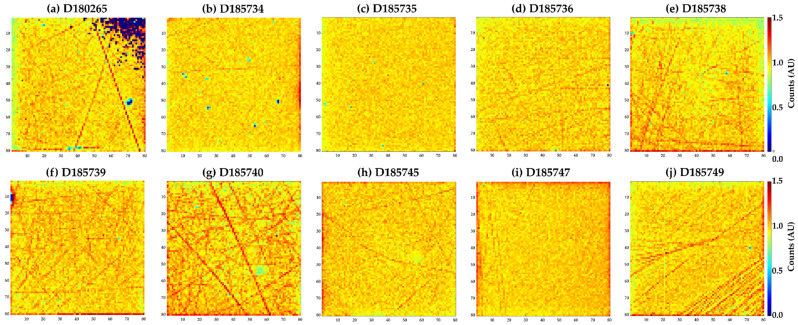
Maps of the intensity per pixel measured for each of the CdZnTe detectors (**a**–**j**) under a flat field irradiation with an ^241^Am-sealed source. The maps were normalized to the median number of counts detected in each detector. The pixel to pixel variations in counts were in the range 5.5–12.2%.

**Figure 7 sensors-20-02747-f007:**
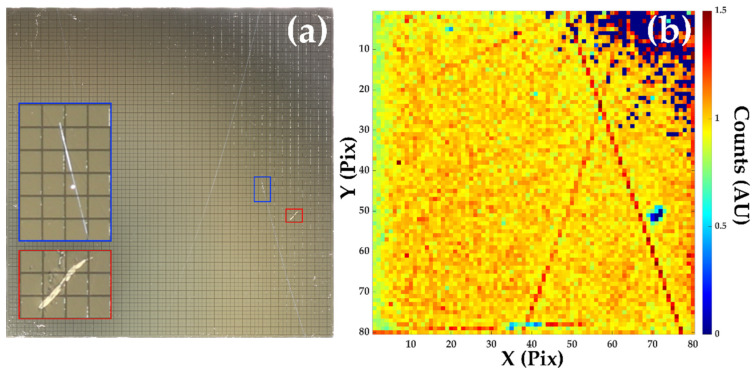
A comparison of an optical image of the 80 × 80, 250 μm pitch, pixels of sensor D180265 prior to hybridization (**a**), and the intensity map produced by a flat field X-ray exposure of the detector module (**b**). The total area of the sensor is 20.35 mm × 20.45 mm. Some correlation between defects in the optical and X-ray images can be observed.

**Figure 8 sensors-20-02747-f008:**
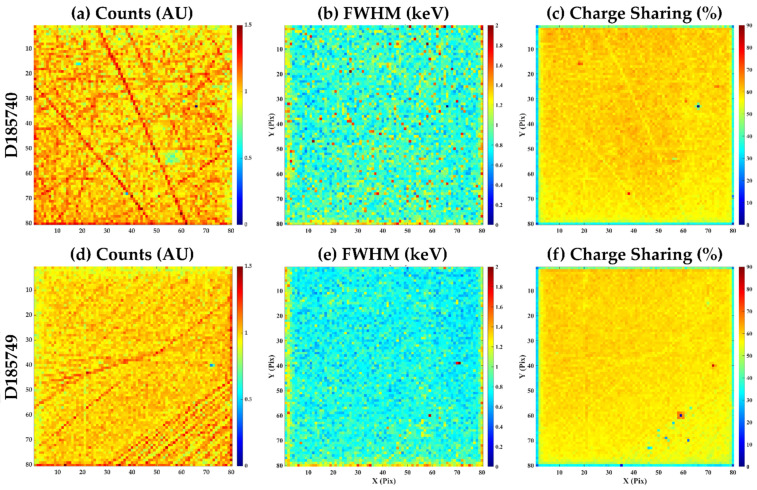
A comparison of the performance of detectors D185740 (**a–c**) and D185749 (**d,e**); the maps show the number of counts per pixel (**a,d**), the FWHM of the 59.54 keV photo peak (**b,e**) and the charge sharing percentage (**column and f**).

**Figure 9 sensors-20-02747-f009:**
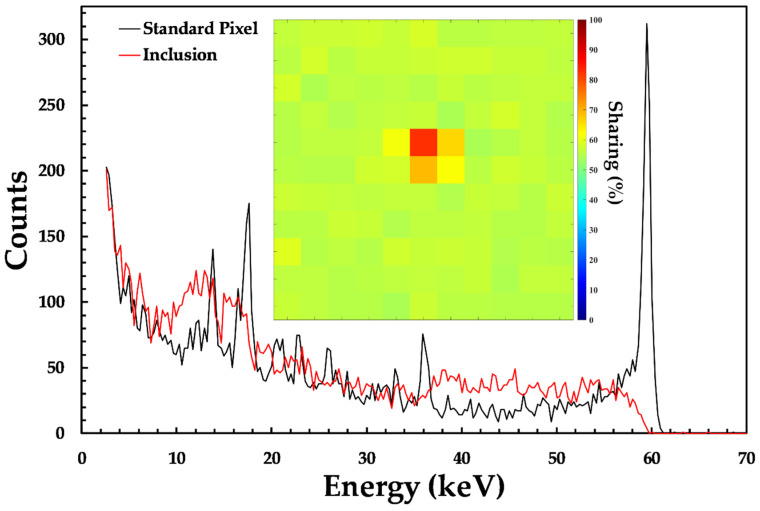
A comparison of the spectroscopic performance of a standard pixel (black solid line) in detector D185734 and a pixel in the same device that is suspected to contain a tellurium inclusion within its volume; (**inset**) a map of the charge sharing percentage measured in the area surrounding the inclusion.

**Figure 10 sensors-20-02747-f010:**
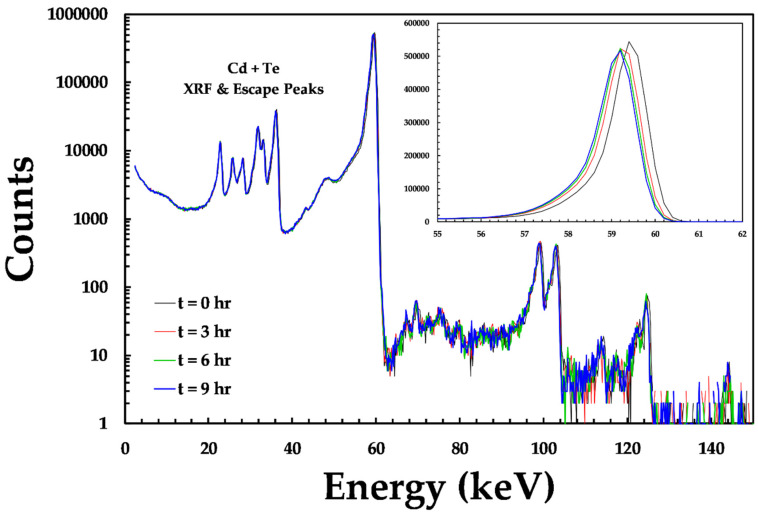
The variation in the spectroscopic performance of detector D185735 as a function of exposure time. (**inset**) the small changes observed in the 59.54 keV photo peak.

**Figure 11 sensors-20-02747-f011:**
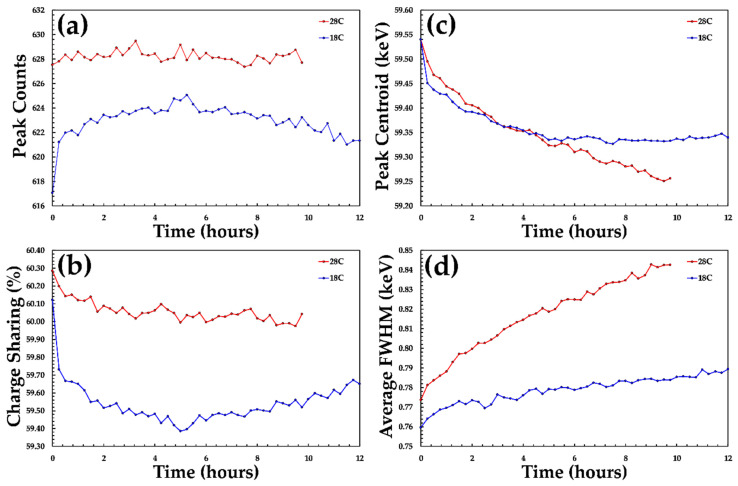
The stability of D185735 over a period of 12 h. Key indicators of the detectors spectroscopic performance are shown. (**a**) the average number of counts detected per pixel; (**b**) the average charge sharing percentage per pixel; (**c**) the centroid of the 59.54 keV photo-peak and (**d**) the average FWHM of the 59.54 keV photo-peak. Data are shown at a temperature of 28 (red) and 18 °C (blue).

**Table 1 sensors-20-02747-t001:** A summary of the spectroscopic performance of the 10 Redlen HF-CdZnTe HEXITEC devices as measured with the 59.54 keV ^241^Am γ-ray. The error in the FWHM value is the standard deviation measured across the 6084 bulk pixels.

Serial Number	Pixel Yield (%)	σ_counts_ (%)	FWHM (keV)	Charge Sharing (%)
D180265	95.52	8.2	0.86 ± 0.22	56.8
D185734	99.93	5.9	0.79 ± 0.14	55.1
D185735	99.93	5.5	0.79 ± 0.15	55.9
D185736	99.98	6.6	0.87 ± 0.17	62.2
D185738	99.92	9.0	0.85 ± 0.17	60.0
D185739	99.98	8.7	0.84 ± 0.13	56.1
D185740	99.93	12.2	0.89 ± 0.19	59.1
D185745	99.93	7.1	0.78 ± 0.13	56.5
D186747	99.89	5.7	0.82 ± 0.16	61.3
D185749	99.98	7.7	0.80 ± 0.13	56.6
**AVERAGE**	**99.50**	**7.7**	**0.83**	**58.0**

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
