# Peer review of "Characterization of the Uniformity of High-Flux CdZnTe Material"

_sensors, 2020, doi:10.3390/s20102747_

Round 1

Reviewer 1 Report

This is a very important paper for the radiation detection, sensors and photonics community. It is well written with very good experimental procedures and results analysis. The 1.39% FWHM obtained for the 59.54-keV gamma line of Am-241 is a very good resolution. Also, observing no significant changes in a 12-hour continuous operation period is very good; the absence of polarization makes this detector very useful in security applications and in high photon flux applications.

I have just a few very minor suggestions:

  1. The scale on the magnified insert image in Figure 1b is not legible. I suggest stating the scale or the dimension of the insert image in the figure caption.
  2. I also suggest adding a scale or stating the image dimension in the caption of Figure 7 (left). The size, and configuration, were stated in subsection 2.1, but it is good to also have it in the figure or figure-caption.
  3. I suggest adding labels to the axes and color codes in Figures 2 (Left), 7 (Right), 8 (left column), and 9 (insert).

Author Response

Dear Reviewer,

Thank you for taking the time to review our manuscript. We appreciate your kind comments and have made the corrections you suggested as detailed below.

Many Thanks,

Matt

Comment 1

As suggested additional comment regarding the scale has been added to the caption of Figure 1.

"(inset) a higher magnification image of a 1.25 mm × 1.75 mm region of the detector after the deposition of silver-loaded-epoxy bumps during hybridization"

Comment 2

As suggested I have modified the caption to give more information on the detector geometry and size of Figure 7.

"Figure 7. A comparison of an optical image of the 80 × 80, 250 mm pitch, pixels of sensor D180265 prior to hybridization (left) and the intensity map produced by a flat field X-ray exposure of the detector module (right). The total area of the sensor is 20.35 mm × 20.45 mm. Some correlation between defects in the optical and X-ray images can be observed."

Comment 3

Corrections have been made as suggested.

Figure 2: Labels to these axis have been labelled. In order for these to be legible I have reshaped the Figure slightly.

Figure 7: Axis labels have been added.

Figure 8: Label updated "Counts (AU)"

Figure 9: Inset label increased in size.

Reviewer 2 Report

The article “Characterization of the uniformity of high flux CdZnTe material” deals with the characterization of CdZnTe detectors with potential application in X-Ray imaging cameras for synchrotron facilities.

The authors characterized the capabilities of the detectors with high professionalism. My comments are regarding mostly the bibliography, which is poor with only 29 citations, out of which 19 are self-citations. This makes the article look more as a presentation for a commercial product than as a scientific article.

As the authors mention, there is a lot of work regarding the production and the study of CdZnTe and CdTe. For the quality of the work to be improved, I find it very important to compare your results to the ones discussed in the literature. As a general observation, some statements are made that are not supported by bibliography and that are not further justified.  

For instance, what do the values of the FWHM mean? Where are they in comparison to other studies?

What does this stability mean in the general (“universal”) picture? How long are the sensors operating usually? How long are the normal tests?    

Also, I think is necessary you add some phrases in the Materials and Methods part about how you obtained the CdZnTe material, before talking about the testing procedure.

Author Response

Dear Reviewer,

First off may I offer my thanks for the time taken reviewing our manuscript, your comments have been very helpful. I've tried to address your comments as detailed below and I hope this has improved the clarity of the paper.

Many Thanks,

Matt

  1. Additional Comment on FWHM

I've now extended Section 3.1 to provide more background on the significance of the FWHM values as well as including references to the work of other groups who are developing CdTe/CdZnTe pixel detectors. A copy of the additional text and references is below:

One key indicator of the performance of compound semiconductor materials such as CdZnTe is their energy resolution, characterized here by the full width at half maximum (FWHM). The energy resolution of a material is dependent on a number of factors including the charge transport properties of the material, the charge carrier mobility (m) and the carrier lifetime (t), as well as the detector geometry and electronics. The magnitude of spatial variations in the energy resolution also provides an excellent indicator of the uniformity of the material.

The ultimate limit of the energy resolution of direct conversion detectors such as CdZnTe are described by the Fano limited energy resolution. When an X-ray is absorbed in the semiconductor the processes that give rise to individual charge carriers are not independent as the band structure of the material limits the numbers of ways that the material may be ionized; as a result the energy resolution that is achievable is better than that predicted by simple statistical considerations. The Fano limited energy resolution (DEFano) is described by Equation 1:

ΔEFano = 2.35 SQRT(FWEγ) Equation 1

Where F is Fano Factor and W is the electron hole pair creation energy these have values of 0.1 and 4.62 eV ehp-1 in CdZnTe material [18]. In recent years the spectroscopic performance of CdZnTe and CdTe detectors have improved dramatically with FWHM measured at 59.54 keV of the order 1 – 2 keV [16, 19-21] compared to the theoretical Fano limited minimum of 0.39 keV.

Maier, D., Blondel, C., Delisle, C., Limousin, O., Martignac, J., Meuris, A., Visticot, F., Daniel, G., Bausson, P-A., Gevin, O., Amoyal, G., Carrel, F., Schoepff, V., Mahe, C., Soufflet, F. and Vassal, M-C., Second generation of portable gamma camera based on Caliste CdTe hybrid technology., Inst. Meth. Phys. Res. A. 2018, Volume 892, pp. 106 – 113. (doi: 10.1016/j.nima.2017.12.027).

Zambon, P., Radicci, V., Trueb, P. Disch, C., Rissi, M., Sakhelashvili, T., Schneebeli, M. and Broennimann, C., Spectral response characterization of CdTe sensors of different pixel size with the IBEX ASIC., Inst. Meth. Phys. Res. A. 2018, Volume 912, pp. 338 – 342. (doi: 10.1016/j.nima.2018.03.006).

Watanabe, S., Ishikawa, S-n., Aono, H., Takeda, S., Odaka, H., Kokubun, M., Takahashi, T., Nakazawa, K., Tajima, H., Onishi, M. and Kuroda, Y., High Energy Resolution Hard X-Ray and Gamma-Ray Imagers Using CdTe Diode Devices., IEEE Trans. Nuc. Sci., 2009, Volume 56:3, pp. 777-782. (doi: 1109/TNS.2008.2008806).

2. Additional Comments on Stability

Based on your suggestions I have included significantly more information on the phenomena that have typically affected the temporal stability of Cd(Zn)Te detectors. I have also included additional information on the requirements of some of the target applications of these detectors. Additional references have been added to support the additional information. Below is the additional content and references:

Another important consideration in accessing the performance of the HF-CdZnTe material is its temporal stability. For techniques like Computed Tomography that are widely used for materials science, security and medical imaging, detectors may be exposed to high fluxes of X-rays for many hours at a time and must remain stable for the duration of the measurements. Historically, temporal variations in the performance of CdTe and CdZnTe detectors have been commonly reported. These phenomena are typically caused by flux-induced [2, 33] and bias-induced polarization [34, 35] in which a buildup of space charge, caused by the trapping of carriers, leads to a progressive modification of the electric field. These changes in electric field result in a reduction in both the counting efficiency and the spectroscopic performance of detectors. The rate of these changes are dependent on a number of factors such as the photon flux, detector electrode type, the bias voltage and the detector temperature.

In the case of bias-induced polarization, the use of a blocking (Schottky) electrode prevents carriers leaving the sensor resulting in the formation of space charge under the electrode independent of the photon flux. The presence of bias-induced polarization is normally addressed through the use of a bias refresh scheme where the detector bias is periodically set to 0 V for a short time of the order of a second in order to allow the space charge to recombine. The period of bias refreshing can be on the order of minutes or hours depending on the detector conditions [16, 34-36]. In the case of flux-induced polarization, the large number of photo-generated carriers results in excessive numbers of trapped holes that form the space charge within the sensor. Once polarization occurs, only a large reduction or removal of the photon flux will allow the detector to recover [2, 33].

 Recent encouraging results from the characterisation of HF-CdZnTe material coupled to single photon counting [7] and integrating detector ASICs [6] have demonstrated that the material is capable of operating at high photon fluxes with negligible changes detected in detector performance. These results suggest that the polarization phenomena in this material is reduced or suppressed. The advantage of the spectroscopic information provided by the HEXITEC ASIC is that it allows any small changes in the detector material under extended operation to be studied; it is worth noting that this is limited to low X-ray fluxes (<105 ph s-1 mm-2). In order to study the detector stability in more detail, data was collected for 300 s at 900 s intervals for a total of 9 – 12 hours with detector D185735. Measurements were made with the detector temperature fixed at the standard 28oC operating temperature as well as at a reduced temperature of 18oC. Figure 10 shows the evolution of the global spectrum at 28oC over the course of a 9 hour exposure. The change in the number of counts, position and width of the 59.54 keV photo-peak as well as the percentage of charge sharing events were evaluated for each acquisition, see Figure 11.

At both of the operating temperatures measured small changes in the overall performance of the detector are observable, however, the changes that are measured are very small. For example, the average FWHM measured in the detector over 9 hours was found to change from 770 eV to 840 eV at 28oC and 760 eV to 780 eV at 18oC. These level of changes are negligible and show no clear spatial correlation within the detector. As with previous studies at high photon fluxes, this suggests that the response of the HF-CdZnTe material shows a high degree of stability which is more than sufficient for the typical operation conditions of these detectors.

Prokesch, M., Soldner, S. A., Sundaram, A. G., Reed, M. D., Li, H., Eger, J. F., Reiber, J. L., Shanor, C. L., Wray, C. L., Emerick, A. J., Peters, A. F. and Jones, C. L., CdZnTe Detectors Operating at X-ray Fluxes of 100 Million Photons/(mm2.sec)., Nuc. Sci. 2016, Volume 63:3, pp. 1854 – 1859. (doi: 10.1109/TNS.2016.2556318).

Cola, A. and Farella, I., The polarization mechanism in CdTe Schottky detectors., Phys. Lett. 2009, Volume 94, 102113. (doi: 10.1063/1.3099051).

Astromskas, V., Gimenez, E. N., Lohstroh, A. and Tartoni, N., Evaluation of Polarization effects of e- Collection Schottky CdTe Medipix3RX Hybrid Pixel Detector., Nuc. Sci. 2016, Volume 63:1, pp. 252 – 258. (doi: 10.1109/TNS.2016.2516827).

Gimenez, E. N., Astomskas, V., Horswell, I., Omar, D., Spiers, J. and Tartoni, N., Development of a Schottky CdTe Medipix3RX hybrid photon counting detector with spatial and energy resolving capabilities., Inst. Meth. Phys. Res. A. 2016, Volume 824, pp. 101 – 103. (doi: 10.1016/j.nima.2015.10.092).

3. Materials and Methods

In this section I have included some additional information on the differences between this type of CdZnTe material (high-flux-grade) to that which has traditionally been used in the community (spectroscopy-grade). I'm afraid that this paticular CdZnTe material has only been available to the academic community for a short amount of time. The only published work on this material that I am aware of are from my group and that of the detector group at ESRF which are already included. The additional text is below:

In spectroscopic-grade CdZnTe material the growth process is optimised to deliver crystals with excellent electron charge transport properties which is ideal for thick sensors with large pixels that are typically used for g-ray spectroscopy applications [3]. The disadvantage of this material is that the hole carrier lifetime is normally very short (~ 0.2 ms) and leads to significant numbers of carriers being trapped. At high photon fluxes the build-up of these trapped holes are the source of the polarization phenomenon. In the HF-CdZnTe material a different approach is taken where the hole lifetime is optimised leading to an increase of an order of magnitude (~ 2.0 ms) [5]. The increase in the hole lifetime enables detectors produced from HF-CdZnTe material to operate at the higher photon fluxes used in many photon science applications.

In my materials and methods section I also state:

In this paper results are reported for the characterisation of a sample of 10 pixelated HF-CdZnTe detectors, each with an area of 4.2 cm2, that have been fabricated by Redlen Technologies and are read out with the STFC’s HEXITEC ASIC [10]

and

The HF-CdZnTe material used in this study was grown by Redlen Technologies using their proprietary THM growth process [4, 5]. Redlen were also responsible for the processing and fabrication of the sensor electrodes for these detectors.

I hope that this makes clear that the CdZnTe sensors were produced solely by Redlen Technologies and that this includes the growth of the material and the fabrication of the detector electrodes.

Reviewer 3 Report

It would be great if authors can write the hypothesis in the Introduction in a more quantifiable manner and add a couple of sentences in conclusion.

Change if any sentence starts with “ This.” Pronouns such as he, she, they, that, and this are words that stand in for other nouns. When using pronouns, however, it is crucial to make it clear which nouns they are standing in for. When the original noun (called the antecedent) appears several sentences before the pronoun that replaces it or does not appear at all, readers may not understand who or what the pronoun refers to.

Place a comma before and.

Specific comments:

Line 35: “ in excess of” replace it with “ over.”

Line 36: Instead of has it should be “ have.”

Line 43: need.

Line 52: It should be material’s.

Line 114: It should be fluorescence.

Line 149: It should be detector’s.

Line 213: Instead of simply use merely.

Line 215: It should be “phenomenon.”

Author Response

Dear Reviewer,

Thank you very much for the time you have taken to review our manuscript, we really appreciate this. I have been through the manuscript and addressed your comments as suggested.

The only comment I have not addressed is the one referring to the use of a serial, or Oxford, comma. My understanding is that this is a personal stylistic choice rather than a grammatical error. British English allows list constructions with or without the Oxford comma.

Many Thanks,

Matt